# Nutritional approach based in self-compassion *versus* energy-restricted diet approach in body dissatisfaction and disordered eating in adult women: a protocol for randomized clinical trial

Alessandra Behar Ramos[1], Vinícius Suedekum da Silva[1], Débora Viçosa Cardoso[1], Carolina Guerini de Souza[1,2,*¤]

1 Postgraduate Program in Food, Nutrition and Health – Universidade Federal do Rio Grande do Sul, Porto Alegre, Rio Grande do Sul, Brazil, 2 Nutrition and Dietetics Division - Hospital de Clínicas de Porto Alegre, Porto Alegre, Rio Grande do Sul, Brazil

¤ Current Address: Postgraduate Program in Food, Nutrition and Health - Universidade Federal do Rio Grande do Sul, Rua Ramiro Barcelos 2400 - 2° andar, Santa Cecília, Porto Alegre, Rio Grande do Sul, Brasil

* carolina.guerini@ufrgs.br

## Abstract

The proposed study protocol aims to compare the effect of a nutritional approach based on self-compassion techniques compared to the traditional method through dieting on body image dissatisfaction, caloric restriction, and dysfunctional eating in women who feel dissatisfied with their bodies.

## Materials and methods

This protocol was developed according to the guidelines of the Standard Protocol Items: Recommendations for Interventional Trials 2013 Statement (SPIRIT) and presents a randomized clinical trial with women who are dissatisfied with their bodies and live in southern Brazil. Participants will be randomized to the self-compassion or diet group and attend eight-week weekly meetings. Body image dissatisfaction will be assessed using the Body Shape Questionnaire (BSQ), dysfunctional eating will be assessed using the Three Factor Eating Questionnaire (TFEQ-RS21), and levels of self-compassion will be measured using the Self-Compassion Scale (SCS). The participants will answer the tools at the beginning of the study, at the end of the 8-week intervention, and 3 and 6 months after the end of the meetings.

## Discussion

This will be the first study to compare these two approaches using body dissatisfaction and eating behavior as outcomes, not just body weight. To date, only

**Data availability statement:** No datasets were generated or analysed during the current study. All relevant data from this study will be made available on the Fighsare platform upon study completion.

**Funding:** Initials of the authors who received each award : CGS Grant numbers awarded to each author: this grant has no nyumber The full name of each funder : Fundação Coordenação de Aperfeiçoamento de Pessoal de Nível Superior (PROAP UFRGS) URL of each funder website: https://www.gov.br/capes/pt-br Did the sponsors or funders play any role in the study design, data collection and analysis, decision to publish, or preparation of the manuscript? No.

**Competing interests:** The authors have declared that no competing interests exist.

observational studies have evaluated the relationship between self-compassion, body dissatisfaction, and dysfunctional eating behavior.

**Trial Registration:** ClinicalTrials.gov NCT06084260

---

## Introduction

Sociocultural factors greatly influence the female body, so to be considered beautiful, it must be slim, with a greater degree of muscle definition and as little fat as possible [1,2]. This standard of beauty has contributed to an increase in body dissatisfaction and low self-esteem in women [3,4]. Dissatisfaction with the body seems to be starting earlier and earlier, being present as early as childhood, and is more prevalent in women than in men [5–7].

Body dissatisfaction is a risk factor for the incessant search for the ideal body, dysfunctional eating behavior [8], as well as the practice of energy-restricted diets [9,10]. Weight loss programs and energy-restricted diets have been practiced for several years [11,12] to deal with body dissatisfaction. Although they show results in the short and medium term, such programs are ineffective in maintaining weight in the long term [11,12] and can contribute to weight regain [13]. In addition, these programs are considered harmful because they favor dysfunctional behaviors' development involving cognitive restriction, binge eating, uncontrollable desire for food, worry, and guilt when eating [14]. Furthermore, such diets, together with the fear of gaining weight and exaggerated preoccupation with food, calories and physical appearance, can be related to a failure's sense, lack of control over one's own life, decreased self-esteem, guilt, irritability, anxiety, depression [12] and the incidence of eating disorders [15–17].

In recent years, approaches have emerged that aim to promote healthy eating behavior without focusing on losing body weight [18] as an alternative to weight loss programs based on food restriction. Interventions without the use of diets have shown psychological benefits, such as an improvement in general psychological well-being and a reduction in suffering, weight-related stigma and body dissatisfaction [19,20], as well as greater eating competence, intuitive eating [21], a reduction in cognitive restriction and lack of eating control, and long-term weight loss [18].

One of these approaches is self-compassion, a theory derived from Buddhism that has shown a positive association with better mental behavior and health parameters [22]. It is a practice that can be learned, accessible anytime, and may relieve suffering [23]. Being self-compassionate implies being moved by one's own suffering and treating oneself with care and empathy [24]. Individuals with worse mental health tend to have less self-compassion [25] while being more self-compassionate tends to reduce psychopathology [26] and help with emotional regulation [27]. In this sense, one study showed a relationship between a higher self-compassion level and a reduction in negative thoughts related to the body [28], a reduction in eating psychopathology and concerns about body image, which may be an adaptive emotional regulation strategy in eating disorders and body dissatisfaction [29].

Other studies have also shown that self-compassionate individuals seem to worry less about their body weight and are less ashamed of their bodies, as well as having a greater self-image valuing and appreciation [29–31]. However, no intervention studies in the literature evaluate the effectiveness of an approach based on self-compassion techniques focused on food compared to the traditional approach with an energy-restricted diet and its impacts on body dissatisfaction. For this reason, the study's main objective is to compare the effect of a nutritional approach using self-compassion techniques with the traditional diet method over eight weeks on body image dissatisfaction and eating behavior in women dissatisfied with their weight.

## Materials and methods

This study protocol for a randomized clinical trial was developed under the guidelines proposed by the Standard Protocol Items: Recommendations for Interventional Trials 2013 Statement (SPIRIT) (S1 Table) [32]. Fig 1 shows the recruitment, intervention, and evaluation schedule.

### Participants, study setting, and eligibility criteria

The study will be conducted with Brazilian women experiencing weight-related body dissatisfaction. To be included in the study, participants must present body dissatisfaction, be between 25 and 50 years old, have access to a cell phone with communication apps (WhatsApp), and be available to attend face-to-face meetings once a week for eight weeks. Participants will be excluded from the study if they have a diagnosis of depression, mood disorders (anxiety, bipolar disorder, borderline), eating disorders, or a history of suicidal ideation; if they have chronic diseases such as diabetes, kidney disease, cardiovascular and/or neurological diseases; and if they are pregnant or have been pregnant for six months or less; and to conclude, if they are menopausal.

|  | STUDY PERIOD | | | | | | |
|  | Enrolment | Allocation | Post-Allocation | | | Follow-up | |
| TIMEPOINT | Week 0 | Week 0 | Week 1 | Weeks 2-7 | Week 8 | After 3 months | After 6 months |
| **ENROLMENT** | | | | | | | |
| Eligibility screen | x | | | | | | |
| Informed consent | x | | | | | | |
| Allocation | | x | | | | | |
| **INTERVENTIONS** | | | | | | | |
| Self-compassion | | | ———— | ———— | ———— | | |
| Diet | | | ———— | ———— | ———— | | |
| **ASSESSMENTS** | | | | | | | |
| Sociodemographic | | | | | | | |
| Weight | | | x | | x | | |
| TFEQ-RS21 | | | x | | x | x | x |
| BSQ | | | x | | x | x | x |
| SCS | | | x | | x | x | x |

*TFEQ-RS21: The Three Factor Eating Questionnaire - R21. BSQ: Body Shape Questionnaire. SCS: Self-Compassion Scale*

**Fig 1. Schedule of recruitment, intervention and evaluation.**

## Recruitment

Recruitment will take place from March 4, 2024, to March 31, 2026, which is the estimated timeframe to achieve the calculated sample size. Participants will be recruited through posts on the researchers' social networks, and those who express an interest will be contacted for an initial conversation via WhatsApp video call. This assessment will confirm the participant's desire to participate in the study, the inclusion and exclusion criteria, and guidance on the study protocol. After confirming eligibility and signing the Informed Consent Form, the participants will be randomized and assigned to one of the two intervention groups: self-compassion or diet. The groups will take place weekly through face-to-face meetings, which will take place over eight weeks and will be one hour long. The group will be led by nutritionists previously trained in the two approaches. The adherence rate will be controlled, requiring a minimum participation of 75% (6 meetings) to be maintained in the entire study (baseline to eight weeks, three and six months follow-up).

Before the first meeting, all participants will answer an anamnesis and three validated scales for assessing eating behavior and body dissatisfaction, which will be available through the *Google Forms* platform. The same scales will be answered after the eight meetings (the end of the group), three months after the last meeting, and six months after the last meeting. Fig 2 shows the study flow diagram.

## Interventions

Self-Compassion Group: the group with a self-compassion approach will have its own protocol created for the study. This intervention was based on self-compassion exercises proposed in "The mindful self-compassion workbook: a proven way to accept yourself, build inner strength, and thrive" [33]. Table 1 shows the topics covered in each meeting.

Diet group: in the diet group, participants will complete a 24-hour food recall (24HR). They will receive a food plan calculated by a nutritionist using the Estimated Energy Requirement calculation [34] as a reference and adjusted to their routine based on the 24HR. The necessary nutritional adjustments will be made to ensure adequate fiber, macro, and micronutrient intake according to the Dietary Reference Intakes [35] for women. The total energy value will depend on the participant's goal, with a calorie deficit of 300 kcal for those wishing to lose weight and a surplus of between 300–500 kcal for those wishing to gain weight. The macronutrient distribution will be the AMDR of 45–65% carbohydrates, 10–35% proteins, and 20–35% lipids. Table 2 shows the topics covered at each meeting. In the fifth week of the study, the participants will have an individual online consultation to re-evaluate the proposed diet and adjust it if necessary instead of a group meeting.

## Adherence, modifications, and concomitant care

In both groups, the importance of attendance and commitment to the meetings will be discussed. It will also be emphasized that it will be a safe space for listening, welcoming, and confidentiality. Participants will be asked about their feelings and doubts about their study process. The meetings will always occur on the same predetermined day, time, and location. Participants will be notified via smartphone before each meeting as a reminder. To increase adherence and commitment to the study, participants will also be offered activities to practice at home, such as food records, observation, and reflection exercises, or they will be asked to bring materials related to the next meeting (e.g., food packaging).

Adherence to the proposed interventions will be checked at each meeting. However, the difficulty of carrying out what is proposed does not compromise remaining in the group meetings.

Participants cannot change the group they have been allocated to. However, they are free to leave the study as they wish. The meeting protocol for both groups cannot be modified, as this could bias the study.

## Outcomes

**Primary outcomes.** Body dissatisfaction will be assessed using the Body Shape Questionnaire (BSQ), translated to Portuguese [36], which is used to identify concerns about body shape and self-deprecation related to physical shape. It has

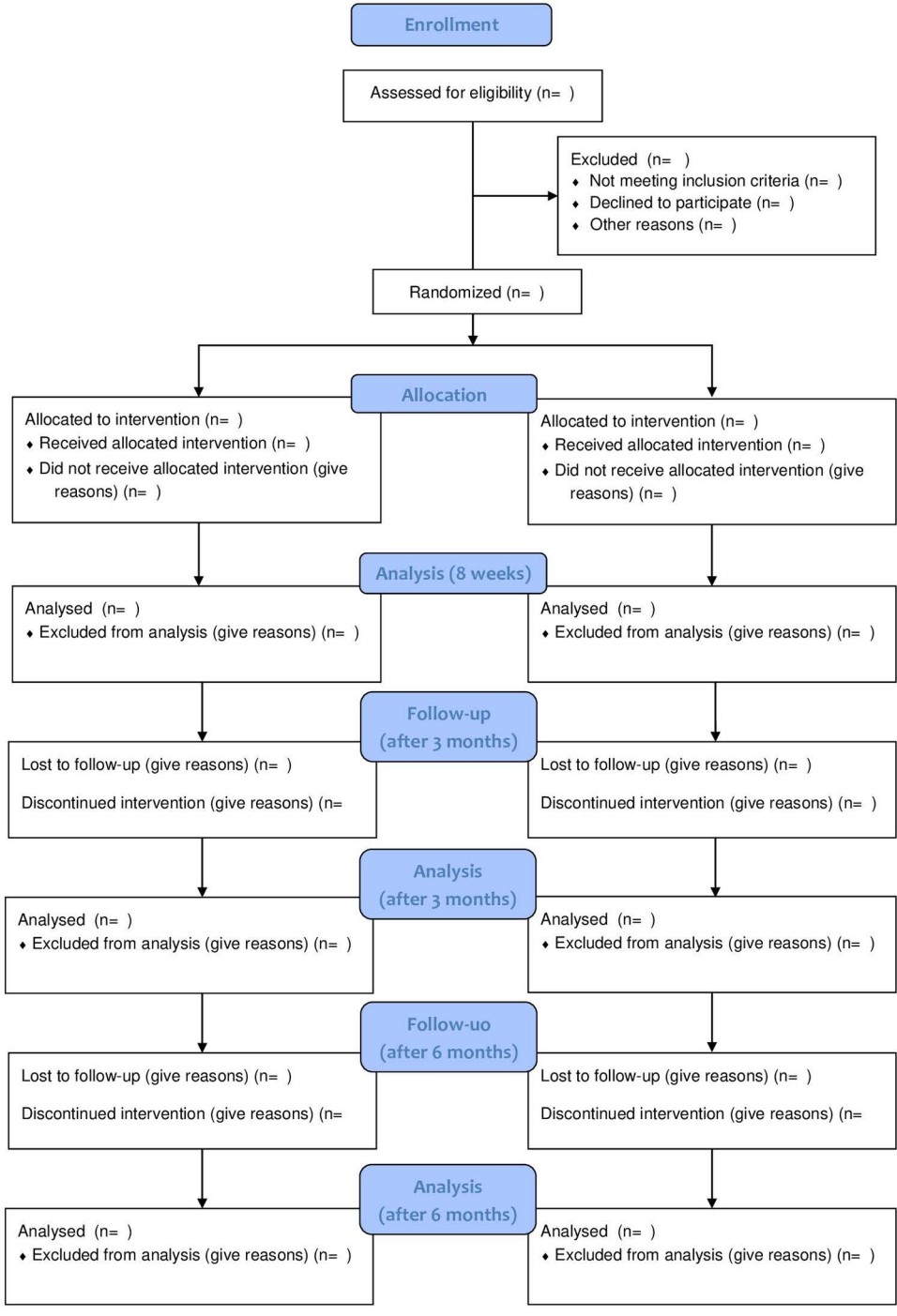

**Fig 2. Flow diagram.**

34 questions related to body image, with a scale from 1 to 6: 1-Never, 2-Rarely, 3-Sometimes, 4-Frequently, 5-Very Often, and 6-Always. The result is obtained by adding up all the scores. A score of less than 80 points means that the individual has no dissatisfaction. Between 80 and 110, there is mild dissatisfaction; between 111 and 140, moderate dissatisfaction, and over 140 severe dissatisfaction.

Dysfunctional eating behavior will be assessed using the instrument "The three-factor eating questionnaire - R21 (TFEQ-R21)" [37], a questionnaire developed to evaluate the eating behavior of obese and eutrophic individuals, translated and validated to Brazilian [38]. The instrument consists of 21 questions, with questions 1–20 having 4 alternatives (scoring from 1 to 4, from the highest severity of dysfunctional eating to the lowest) and the last question having a scale from 1 to 8 (from the lowest degree of food restriction to the highest). Three dimensions of eating behavior are assessed: cognitive restriction, uncontrolled eating and emotional eating. Cognitive restriction refers to a set of self-imposed food obligations and prohibitions in order to lose or maintain weight. However, when there is a limitation on food consumption, both qualitatively and quantitatively, these individuals, in certain situations such as exposure to a forbidden food, lose self-control and tend to overeat with or without the presence of hunger, characterizing uncontrolled eating. Finally, emotional eating refers to emotions influence on food intake, which can lead to less healthy choices. It is an appropriate instrument for its purpose and serves to support an approach that values the behaviors resulting from poor eating, such as external stimuli and emotions [38]. The higher the score, the more dysfunctional the behavior.

**Secondary outcome.** Self-compassion levels: the Self-Compassion Scale (SCS) will be used, with 26 items, which was developed to measure self-compassion in three components: self-judgment versus self-kindness, sense of isolation versus common humanity and hyper-identification versus mindfulness [39]. The items are grouped into six subscales: Self-kindness (items 5, 12, 19, 23, 26); Self-judgment (items 1, 8, 11, 16, 21); Common Humanity (items 3, 7, 10, 15); Isolation (items 4, 13, 18, 25); Mindfulness (items 9, 14, 17, 22) and Over-identification (items 2, 6, 20, 24). Each item is rated on a 5-point Likert scale (1=Almost never; 5=Almost always). The total score is obtained from the average of the six subscales. Higher scores mean more self-compassion.

Sociodemographic characteristics: to gather information on the participants' characteristics, an anamnesis form will be used, in which the participants will fill out questions on sociodemographic issues (name, age, gender, marital status and

**Table 1. Description of the meetings in the Self-Compassion group.**

| Meeting | Central theme | Dynamics |
|---|---|---|
| Meeting 1: How have I dealt with myself when it comes to my body and food? | Presentation and measurement of anthropometric measurements (weight and height) Self-criticism | Reflection on self-criticism |
| Meeting 2: Changing my relationship with my body through self-compassion" | Self-compassion and self-kindness | How do I treat a friend? Mindfulness exercise (Gabi Damasceno) adapted |
| Meeting 3: Where does it all come from? Looking at my history with food and my body | Mindfulness | What would you change about your body"video Accepting our bodies with self-compassion |
| Meeting 4: Shared humanity in body dissatisfaction and food: is it just me? | Shared humanity | Reflection exercise on shared humanity |
| Meeting 5: Mindful eating: the power of observation | Mindful eating | Mindful eating exercise |
| Meeting 6: Changing my eating habits in a compassionate way | Self-compassion as a tool for changing habits | Changing in a self-critical way vs. a self-compassionate way Dynamics in pairs |
| Meeting 7: Changing the way how I look at my body | Mindfulness Self-kindness | Mindfulness exercise (describing vs. judging) Self-compassionate letter to the body 100th birthday party"exercise |
| Meeting 8: Cultivating self-compassion in my life - what I've changed in how I look at my food and body. | Resumption of all previous subjects | My Mad Fat Diary"video scene Dynamics in pairs Closing |

income, education), as well as general health and lifestyle questions such as: physical activity, smoking, alcohol use, psychological support, perceptions about the body and diet and any other information that the participant can provide to better conduct the research.

## Sample size

Considering the study by Albertson, Neff and Dill-Shackleford (2014) [40], which found an improvement in body dissatisfaction from meditation based on self-compassion techniques with an effect size of 0.73 (high), a sample size of 76 people (38 for each group) was calculated. This calculation considered a power of 80%, a significance level of 5% and a Cohen's d of 0.7 to test whether there is a minimal difference in the body dissatisfaction averages and self-compassion levels between the intervention and control groups, already including an addition of 10% for possible losses and refusals.

## Randomization, allocation and masking of groups

Participants will be randomized to the Self-Compassion or Diet group with 1:1 allocation in the Random Group Generator (https://pt.rakko.tools/tools/59/). The nutritionists responsible for applying the interventions will have no contact with the randomization and will not know in advance who will go to which group, only the research coordinator.

Due to the nature of the intervention, it is not possible to blind the researchers and participants during the study's conduction after the randomization process. However, the intervention and control groups will not interact in any way. Likewise, the statistical analysis will be blinded to the evaluator.

## Statistical analysis

Baseline characteristics of the participants will be compared between groups (self-compassion and diet). For categorical variables, the Pearson Chi-square test will be used, with continuity correction for dichotomous variables or Fisher's exact test when at least one cell has an expected count of less than five. For continuous numerical variables, the Shapiro-Wilk test will assess normality. If normality is accepted, results will be presented as mean and standard deviation and analyzed using Student's t-test for independent samples. If the distribution is non-parametric, data will be expressed as median and interquartile range and analyzed using the Mann-Whitney U test. No significant differences between groups at baseline

**Table 2. Description of meetings in the Diet group.**

| Meeting | Central theme | Dynamics |
|---|---|---|
| Meeting 1:<br>Presentation and performance of anthropometric measurements | Presentation and measurement of anthropometric measurements (weight and height)<br>Application of anamnesis to calculate the diet plan | Conversation about the purpose of the group and the participants |
| Meeting 2:<br>Motivation for weight loss | Consequences of being overweight and sedentary and the importance of cultivating healthy lifestyle habits | Written reflection |
| Meeting 3:<br>How to choose food? | Classification of foods according to the Food Guide for the Brazilian population (2014) | Food classification dynamics |
| Meeting 4:<br>How to read food labels? | Reading food labels | "Which food is healthier" dynamic |
| Meeting 5:<br>Guidance and monitoring of the diet plan | Food plan | Individual guidance on the diet plan (online) |
| Meeting 6:<br>Myths and truths about healthy eating | Demystifying beliefs about healthy eating | Dynamics of myths and truths |
| Meeting 7:<br>How to use carbohydrates to my advantage | Glycemic index and glycemic load<br>Smart food combinations | High-fiber foods vs. low-fiber foods |
| Meeting 8:<br>Cultivating healthy habits in my life | Ways to cultivate healthy habits in the long term | Clarification of doubts<br>Closing |

are expected due to the randomization process, which should provide similar groups. However, if significant differences are observed, these variables will be included as adjustments in multivariable analyses, as described below for primary and secondary outcomes.

Comparisons between baseline and follow-up for the primary and secondary outcomes will be tested using a General Linear Model (GLM) for repeated measures, considering the interaction between group (self-compassion and diet) and time (before and after). Separate analyses will be conducted for each follow-up point: from baseline to the end of the 8-week intervention, from baseline to 3 months after the intervention, and from baseline to 6 months after the intervention. Differences will be considered statistically significant at $p < 0.05$. The data will be analyzed using Statistical Package for the Social Sciences (SPSS) software, version 26.0.

## Data management

The data will be collected by two independent authors and stored in an SPSS® file database. The tools used to obtain the outcomes will be evaluated at the beginning (week 0) and after the intervention (week 8) and again at three and six months *follow-up*. All forms will be made available via *Google Forms.*

One of the researchers will send a WhatsApp message every week to promote participant retention and complete follow-up, encouraging participants to come to the meetings. Also, the messages sent will be the same for both groups and will include reminders about the meeting days, location, time, and any materials participants may need to bring for the activities. To those who discontinue or deviate from the intervention protocol, all data exclusion will be necessary.

The data collected will be saved on an institutional drive linked to the researcher in charge and shared only with the other researchers in the study. The data are expected to be stored for two and a half years (until the end of 2025), following the study schedule.

An interim analysis could be conducted if preliminary results need to be presented at scientific events relevant to the institutions involved in the study's development. However, if such analyses could compromise the reliability of the data, they will not be performed. All authors could have access to interim results, but only the study coordinator (CGS) could decide to terminate the trial.

The plans for investigators and sponsor to communicate trial results are: a) Individual disclosure of the basal and post-intervention results to each participant; b) A public seminar with grouped results at sponsor institution to healthcare professionals; c) Public disclosure in scientific events, preprint, and scientific article publication.

## Ethical aspects

The study was conducted in accordance with the Declaration of Helsinki and approved through the Research Ethics Committee from Hospital de Clínicas de Porto Alegre (HCPA) under number 2022–0634, CAAE 67679423.0.0000.5327. All participants will provide written informed consent before data collection.

Any changes to the protocol that could have an impact on the study's conduction, including changes to the objectives, sample size, tools or that affect the participants in any way, will be communicated immediately to the Ethics Committee via email. Participants will be asked to read and sign the Informed Consent Form (ICF) when they enter the study. Participants will be given full details of the study's characteristics and interventions. They will also be informed that they are free to withdraw from the study at any time. The Research Ethics Committee, which approved the study, demands periodical reports for the trial's auditing. This process occurs with the investigators' participation.

Adverse events will be monitored weekly by researchers (ABR, DCV, VSS) in each meeting through conversations with the participants. According to the type of adverse event, referral to the health care service of the sponsor institution will be made. The health care service of the sponsor institution will provide post-trial care and compensation to those who suffer harm for trial participation.

## Discussion

This study proposes a nutritional intervention protocol based on self-compassion techniques compared to a traditional approach with dieting, both for eight weeks, evaluating which has better results in body dissatisfaction and eating behavior in adult women with weight-related body dissatisfaction. Our hypothesis is that the intervention based on self-compassion techniques will be more effective in reducing body dissatisfaction and dysfunctional eating behavior in the participants.

Traditional weight loss programs, or restrictive diets, have been known and practiced for years [11,12]. These models encourage individuals to consciously restrict their diet and are effective in the short and medium term. However, studies indicate that the strategies generate emotional stress and develop dysfunctional behaviors in relation to food in the long term, compromising the individual's physical and mental health [11,12,14]. In this sense, non-dietary interventions have shown important psychological benefits, such as improved well-being and decreased suffering, weight-related stigma, body dissatisfaction [19,20] greater eating competence [21] decreased cognitive restriction and lack of eating control [18]. For this reason, other approaches are needed to address body dissatisfaction issues and the relationship with food in adults.

To date, only observational studies have evaluated the relationship between self-compassion and body dissatisfaction and dysfunctional eating behavior. A study of 435 women, which aimed to relate self-compassion to the pressure for a thin body, indicated that self-compassion decreased the media pressure related to thinness, dysfunctional eating and the internalization of the thin ideal [41]. In women diagnosed with eating disorders who perceive themselves as disgusting and have an aversion to their bodies, self-compassion has also been shown to be positive [30]. More self-compassionate individuals seem to worry less about their bodies and weight and have more body appreciation [42], less body shame [31] and are less at risk of eating disorders and other harmful behaviors [43].

As strengths, it is important to note that this is the first study to compare these two approaches using body dissatisfaction and eating behavior as the outcome, not body weight. The intervention time of eight weeks can also be considered a strength, due to the depth of the intervention that can be made with a longer duration of the study, since similar studies have shorter intervention times, from three to a maximum of five meetings. In addition, it will be observed which approach is more effective in the follow-up after the face-to-face meetings have finished, with the participants answering the questionnaires again after three and six months. Limitations include the impossibility of blinding the study, due to the nature of the intervention, so it is not possible to blind the participants or the professionals who will conduct the meetings. As this is a face-to-face study, it can only include people who can go to the study site, which may limit the sample's diversity and potential for generalization.

## Supporting information

**S1 Table.  SPIRIT checklist.**
(DOCX)

## Acknowledgments

The authors would like to thank the Universidade Federal do Rio Grande do Sul (UFRGS) for academic support, and Professor Vivian Cristine Luft for her assistance with statistical analyses.

## Author contributions

**Conceptualization:** Alessandra Behar Ramos, Carolina Guerini de Souza.

**Data curation:** Alessandra Behar Ramos.

**Investigation:** Alessandra Behar Ramos, Vinícius Suedekum da Silva, Débora Viçosa Cardoso.

**Methodology:** Alessandra Behar Ramos, Vinícius Suedekum da Silva, Carolina Guerini de Souza.

**Resources:** Carolina Guerini de Souza.

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
