## [Decision Letter · Decision Letter 0]

22 Dec 2024

PONE-D-24-40704Nutritional approach based in self-compassion versus energy-restricted diet approach in body dissatisfaction and disordered eating in adult women: a protocol for randomized clinical trial.PLOS ONE

Dear Dr. de Souza,

Thank you for submitting your manuscript to PLOS ONE. After careful consideration, we feel that it has merit but does not fully meet PLOS ONE’s publication criteria as it currently stands. Therefore, we invite you to submit a revised version of the manuscript that addresses the points raised during the review process.

We look forward to receiving your revised manuscript.

Kind regards,

Leonardo Vidal Andreato, PhD

Academic Editor

PLOS ONE

Journal Requirements:

The authors would like to thank the Fundação de Amparo à Pesquisa do Estado do Rio Grande do Sul (FAPERGS), Hospital de Clínicas de Porto Alegre and Universidade Federal do Rio Grande do Sul (UFRGS) for supporting this research.

Initials of the authors who received each award : CGS

Grant numbers awarded to each author: this grant has no nyumber

The full name of each funder :  Fundação Coordenação de Aperfeiçoamento de Pessoal de Nível Superior (PROAP UFRGS)

URL of each funder website: https://www.gov.br/capes/pt-br

Did the sponsors or funders play any role in the study design, data collection and analysis, decision to publish, or preparation of the manuscript? No.

Reviewers' comments:

Reviewer's Responses to Questions

**Comments to the Author**

1. Does the manuscript provide a valid rationale for the proposed study, with clearly identified and justified research questions?

Reviewer #1: Yes

Reviewer #2: Yes

2. Is the protocol technically sound and planned in a manner that will lead to a meaningful outcome and allow testing the stated hypotheses?

Reviewer #1: Partly

Reviewer #2: Partly

3. Is the methodology feasible and described in sufficient detail to allow the work to be replicable?

Reviewer #1: Yes

Reviewer #2: Yes

4. Have the authors described where all data underlying the findings will be made available when the study is complete?

Reviewer #1: Yes

Reviewer #2: Yes

5. Is the manuscript presented in an intelligible fashion and written in standard English?

Reviewer #1: Yes

Reviewer #2: Yes

6. Review Comments to the Author

You may also provide optional suggestions and comments to authors that they might find helpful in planning their study.

Reviewer #1: The authors present a protocol for a randomized trial evaluating self-compassion training versus a diet group among Brazilian women who are dissatisfied with their bodies. Both groups will attend weekly meetings for 8 weeks. Assessments will be made at baseline, 8 weeks, and 3 and 6 months after the end of the meetings. Primary outcomes are body image dissatisfaction and dysfunctional eating, while the secondary outcome will be level of self-compassion. The manuscript will be strengthened if the authors consider the following points.

1. Authors state that the study will be conducted in women who are dissatisfied with their bodies, yet there is no inclusion criterion regarding body dissatisfaction.

2. Authors mention the primary outcomes as body dissatisfaction and dysfunctional eating, but they do not specify the specific time point that will be the primary comparison. I'm guessing that will be after the meetings, so at 8 weeks, but this should be clarified.

3. In the power calculations/sample size justification, authors should specify the settings for the calculations (for example, two-sided or one-sided test, alpha level and power).

4. The statistical analysis section should provide more detail. For example, will authors compare characteristics of the groups at baseline? What will authors do if there are differences between the groups at baseline? Authors should clarify how the models they will use will handle the repeated measures across individuals. There are also 3 "after" time points. Based on Figure 2, it appears as though analyses will be repeated 3 times (at 8 weeks, at 3 months post intervention and at 6 months post intervention). It is not clear if each of those analyses will use baseline and the single follow-up time in the analysis.

5. Authors indicated weekly contact with participants to promote retention. Will that weekly contact continue throughout the follow-up period as well? Authors should indicate what sort of messages will be sent and whether they are the same for both groups.

6. In lines 262-263, authors indicate those who discontinue will be removed from analyses. Authors should clarify. For example, it might make sense to remove someone if they do not complete the 8 weeks of meetings, since all they will have is a baseline evaluation and are not providing any information regarding impact of either intervention. What about someone who drops out after the 8 weeks and doesn't return for the two additional follow-ups?

7. Authors indicate the interim analyses may be conducted to present results at scientific events and that all co-authors will have access to those results. Authors should indicate the timing of such interim analyses or requirements before even thinking about performing an interim analysis. Also, it is problematic if authors conduct an interim analysis with co-authors all seeing the results, since it is not clear if those authors are involved in delivering the intervention or contacting participants. Typically, an interim analysis is not widely shared to preserve an unbiased study team (and unbiased participants).

8. Authors mention that adverse events will be monitored. Will they also be recorded and reported on? Authors should also clarify what they will consider as adverse events (give examples).

Minor points:

1. lines 174-175 - this sentence is awkwardly phrased, so authors should consider rephrasing it or splitting it into two sentences.

2. line 265: "The data is" should be "The data are"

Reviewer #2: The study “Nutritional approach based on self-compassion versus energy-restricted diet approach in body dissatisfaction and disordered eating in adult women: a protocol for a randomized clinical trial” aimed to analyze the effects of a nutritional approach involving techniques that promote self-compassion compared to a traditional approach in women experiencing body image dissatisfaction, dietary restriction, and dysfunctional eating habits. I believe the article addresses an important and well-structured topic; however, I have a few questions discribe in the reviwer attachment.

7. PLOS authors have the option to publish the peer review history of their article (what does this mean? ). If published, this will include your full peer review and any attached files.

**Do you want your identity to be public for this peer review?** For information about this choice, including consent withdrawal, please see our Privacy Policy .

Reviewer #1: No

Reviewer #2: **Yes: ** Gabriel Fassina Ladeia

---

## [Author Response · Author response to Decision Letter 1]

26 Feb 2025

Porto Alegre, Feb 20th 2025

Response to reviewers of PLOS One ID PONE-D-24-40704: "Nutritional approach based in self-compassion versus energy-restricted diet approach in body dissatisfaction and disordered eating in adult women: a protocol for randomized clinical trial."

First, we thank the reviewers for the comments and suggestions that helped improve our manuscript’s quality. All alterations are highlighted in the text.

Reviewer #1: The authors present a protocol for a randomized trial evaluating self-compassion training versus a diet group among Brazilian women who are dissatisfied with their bodies. Both groups will attend weekly meetings for 8 weeks. Assessments will be made at baseline, 8 weeks, and 3 and 6 months after the end of the meetings. Primary outcomes are body image dissatisfaction and dysfunctional eating, while the secondary outcome will be level of self-compassion. The manuscript will be strengthened if the authors consider the following points.

1. Authors state that the study will be conducted in women who are dissatisfied with their bodies, yet there is no inclusion criterion regarding body dissatisfaction.

Response: We appreciate the observation. We have included body dissatisfaction as part of the inclusion criteria in the Participants, study setting, and eligibility criteria section.

2. Authors mention the primary outcomes as body dissatisfaction and dysfunctional eating, but they do not specify the specific time point that will be the primary comparison. I'm guessing that will be after the meetings, so at 8 weeks, but this should be clarified.

Response: Our intention is to compare the participants at baseline, after the 8 weeks, and to repeat the analyses 3 and 6 months after the intervention. This information is now highlighted at the end of Recruitment Section.

3. In the power calculations/sample size justification, authors should specify the settings for the calculations (for example, two-sided or one-sided test, alpha level and power).

Response: Thank you for this important observation. We have updated the information in the Sample Size section.

4. The statistical analysis section should provide more detail. For example, will authors compare characteristics of the groups at baseline? What will authors do if there are differences between the groups at baseline? Authors should clarify how the models they will use will handle the repeated measures across individuals. There are also 3 "after" time points. Based on Figure 2, it appears as though analyses will be repeated 3 times (at 8 weeks, at 3 months post intervention and at 6 months post intervention). It is not clear if each of those analyses will use baseline and the single follow-up time in the analysis.

Response: Once again, thank you for this valuable observation. The description of the statistical analysis has been improved, and you can find it in the section with the same title.

5. Authors indicated weekly contact with participants to promote retention. Will that weekly contact continue throughout the follow-up period as well? Authors should indicate what sort of messages will be sent and whether they are the same for both groups.

Response: We appreciate your observation. The messages will be the same for both groups and sent by WhatsApp. These messages will include reminders about the meeting days, location, time, and any materials participants may need to bring for the activities, according to the group they belong to. During the follow-up period, there will be no further contact, as we intend to assess the effectiveness of sustaining the practices without researcher supervision. We have included this information in the Data Management section.

6. In lines 262-263, authors indicate those who discontinue will be removed from analyses. Authors should clarify. For example, it might make sense to remove someone if they do not complete the 8 weeks of meetings, since all they will have is a baseline evaluation and are not providing any information regarding impact of either intervention. What about someone who drops out after the 8 weeks and doesn't return for the two additional follow-ups?

Response: A minimum attendance of 75% (6 meetings) will be required for participants to remain in the study. If a participant completes all the meetings but does not participate in the follow-up, their data will still be included in the analysis of the 8 weeks compared to the baseline. We inserted more information about this in Recruitment Section.

7. Authors indicate the interim analyses may be conducted to present results at scientific events and that all co-authors will have access to those results. Authors should indicate the timing of such interim analyses or requirements before even thinking about performing an interim analysis. Also, it is problematic if authors conduct an interim analysis with co-authors all seeing the results, since it is not clear if those authors are involved in delivering the intervention or contacting participants. Typically, an interim analysis is not widely shared to preserve an unbiased study team (and unbiased participants).

Response: We appreciate this observation. Interim analyses could be conducted if preliminary results need to be presented at scientific events relevant to the institutions involved in the study's development. However, if such analyses could compromise the reliability of the data, they will not be performed.

Study collaborators already have access to the individual results of each participant, as they are responsible for carrying out the interventions and collecting the data. One factor that minimizes measurement bias is that the data are self-reported. In other words, study collaborators do not influence the responses, as these are standardized according to the assessment instruments used in the study, which are completed online, separate from the intervention sessions. Therefore, the final outcome of the instrument application is entirely at the participant's discretion. The evaluator performing the statistical analyses is fully blinded, not involved in the intervention or data collection, and works exclusively with the fully anonymized database.Adittional information was inserted in

8. Authors mention that adverse events will be monitored. Will they also be recorded and reported on? Authors should also clarify what they will consider as adverse events (give examples).

Response: Thank you for your observation and concern. We hypothesize that participants in the diet group may experience gastrointestinal changes, such as flatulence, due to the increased fiber intake. As for participants in the self-compassion group, psychological discomfort related to body dissatisfaction, eating dysfunctions, or self-esteem issues may arise. Any cases requiring professional intervention will be referred to the public health system. All types of discomfort are monitored during each weekly meeting, documented, and reported biannually to the Ethics Committee.

Minor points:

1. lines 174-175 - this sentence is awkwardly phrased, so authors should consider rephrasing it or splitting it into two sentences.

Response: We appreciate your suggestion and have revised the sentences accordingly. You can find the changes in Adherence, modifications, and concomitant care Section.

2. line 265: "The data is" should be "The data are"

Response: Thank you for the correction. We have made the necessary changes to the text, and you can see them in the Data Management section.

Reviewer #2: The study “Nutritional approach based on self-compassion versus energy-restricted diet approach in body dissatisfaction and disordered eating in adult women: a protocol for a randomized clinical trial” aimed to analyze the effects of a nutritional approach involving techniques that promote self-compassion compared to a traditional approach in women experiencing body image dissatisfaction, dietary restriction, and dysfunctional eating habits. I believe the article addresses an important and well-structured topic; however, I have a few questions discribe in the reviwer attachment.

[Page 2 - Line 33]

Food restriction

Would this be the most appropriate term? What about using caloric restriction or caloric deficit instead?

Response: We appreciate your suggestion and changed the term to caloric restriction.

[Page 2 - Line 44]

“This will be the first study to compare these two approaches using body dissatisfaction and eating behavior as outcomes, not just body weight.”

In the discussion, it seems to justify the reasoning behind the authors' decision to conduct the study.

Response: We understand your point and will provide clarification. In the Discussion section, we aim to emphasize the central focus of the study: the comparison of two very different approaches. We will highlight two outcomes that are not typically prioritized, as body weight is often the primary focus in studies involving caloric restriction. With this statement, we intend to convey that our comparison of these two approaches goes beyond body weight, with a focus on the components of eating behavior;

[Page 3 - Line 60]

“Although they show results in the short and medium term, such programs are ineffective in maintaining weight in the long term and can contribute to weight regain.”

It seems to imply that the programs are ineffective.

Response: Thank you for the observation. Our searches have shown that weight loss programs through diets are effective for weight loss. However, studies indicate that, in the long term, these strategies do not maintain the same level of effectiveness and may contribute to regaining the lost weight. Our objective is not to compare which strategy is more effective for weight loss, but rather to assess their impact on improving body dissatisfaction.

[Page 4 - Line 78]

“It is a practice that can be learned, accessible anytime, and may relieve suffering [23].”

The meditation practice requires time to master, yet here it gives the impression of being "easy."

Response: We appreciate the observation and agree that meditation is a complex technique that takes time to master. However, the text refers to the practice of self-compassion and not meditation.

[Page 5 - Line 107]

“Be excluded from the study if they have a diagnosis of depression, mood disorders (anxiety, bipolar disorder, borderline), eating disorders, or a history of suicidal ideation;”

Wouldn’t this sample be too restrictive? This exclusion criterion omits a broad group, often associated with body dissatisfaction.

Response: We appreciate your comment and have previously discussed this same concern. Although body dissatisfaction is related to the factors mentioned, we chose to exclude such diagnoses to avoid any potential bias associated with these factors and to focus solely on analyzing the intervention’s impact on body dissatisfaction. This decision was made to prevent a selection bias.

[Page 5 - Line 116]

“Conversation via WhatsApp video call.”

Would it not be better to use Google Forms for registration to avoid needing to "disconnect" participants from the study? Calling each participant would be more time-consuming.

Response: We understand that calling each interested volunteer requires more time. However, we believe it is the best way to clarify how each stage of the study will be conducted and to align expectations to minimize dropouts.

[Page 7 - Line 144]

“24-hour food recall (24HR).”

Using this questionnaire might lead to discrepancies in dietary intake reports, as one patient could respond on a Monday while another respond on a Wednesday.

Response: We understand your concern and would like to clarify: the food recall will be completed by the participants during the first meeting of the study only used to gain insight into the participants' eating habits and assist in creating a dietary plan that is more closely aligned with their reality.

[Page 8 - Line 149]

“The total energy value will depend on the participant's goal, with a calorie deficit of 300 kcal for those wishing to lose weight and a surplus of between 300–500 kcal for those wishing to gain weight.”

What criteria were used to determine the 300 kcal deficit? And for the 300–500 kcal surplus? How will this be implemented?

Response: Thank you for your question. According to the “Clinical Protocol and Therapeutic Guidelines (PCDT) for Overweight and Obesity in Adults,” a guideline issued by the Ministry of Health of Brazil (Brazil, 2022), "the prescription of a diet that restricts 500 to 1,000 kcal/day from estimated energy expenditure has been shown to be effective in reducing body weight in overweight or obese individuals and can be recommended in the dietary plan." The same document also states, “A more flexible eating plan aimed at gradual changes is generally more successful. In general, individuals should be advised on how to strategically reduce their intake of certain foods within a healthy, appealing, and convenient eating pattern, tailored to their individual realities and cultural preferences.”

Based on the first premise of the guideline, we initially decided on a caloric restriction of 500 kcal. However, taking into account previous experiences from our research group and the advice provided by the second premise, we chose to reduce the restriction to 300 kcal in order to minimize the risk of non-adherence to the higher caloric restriction. This decision was also influenced by the fact that weight loss is not our primary or secondary outcome.

To implement the caloric restriction, the participant's energy needs are first estimated, and then 300 kcal are subtracted from this value. The meal plan is then calculated based on this adjusted energy requirement. For weight gain, after determining the energy estimate, 300 to 500 kcal are added, depending on the tolerance reported by the participant during the initial anamnesis conducted before the intervention begins.

1https://www.gov.br/saude/pt-br/composicao/saps/ecv/publicacoes/protocolo-clinico-e-diretrizes-terapeuticas-pcdt-para-sobrepeso-e-obesidade-em-adultos/view

[Page 10 - Line 177]

“Before each meeting, participants will be notified via smartphone about the day, time, and place of the next meeting.”

Would not using pre-scheduled dates for meetings increase the dropout rate?

Response: We appreciate the comment. The meetings will have predetermined days, times and locations, and these notifications will only be used as reminders to prevent participants from missing them due to possible forgetfulness. We have revised the sentence in the Adherence, modifications, and concomitant care Section to ensure better clarity.

[Page 10 - Line 191]

“Translated into Brazilian.”

Would “Translate to Portuguese” be a more appropriate term?

Response: Thank you for the observation. We have made the necessary changes to the sentence.

[Page 12 - Line 240]

“Random Group Generator (https://pt.rakko.tools/tools/59/).”

Is it necessary to include the link? It could make the text appear "cluttered."

Response: We understand your point. However, we believe that including the link is important to maintain transparency in the randomization process.

[Page 14 - Line 267]

“An interim analysis could be performed to present the study's preliminary results in scientific events. All authors could have access to interim results, but only the study coordinator (CGS) could decide to terminate the trial.”

Is it necessary to mention this?

Response: We are grateful for the observation. We designed our entire protocol based on the Standard Protocol Items: Recommendations for Interventional Trials (SPIRIT), which includes and requires explanations regarding a possible interim analysis. We thought mentioning it was best to ensure all items are addressed.

---

## [Decision Letter · Decision Letter 1]

21 Apr 2025

Nutritional approach based in self-compassion versus energy-restricted diet approach in body dissatisfaction and disordered eating in adult women: a protocol for randomized clinical trial.

PONE-D-24-40704R1

Dear Dr. de Souza,

We’re pleased to inform you that your manuscript has been judged scientifically suitable for publication and will be formally accepted for publication once it meets all outstanding technical requirements.

Kind regards,

Leonardo Vidal Andreato, PhD

Academic Editor

PLOS ONE

Additional Editor Comments (optional):

Reviewers' comments:

Reviewer's Responses to Questions

**Comments to the Author**

1. Does the manuscript provide a valid rationale for the proposed study, with clearly identified and justified research questions?

Reviewer #1: Yes

Reviewer #2: Yes

2. Is the protocol technically sound and planned in a manner that will lead to a meaningful outcome and allow testing the stated hypotheses?

Reviewer #1: Yes

Reviewer #2: Yes

3. Is the methodology feasible and described in sufficient detail to allow the work to be replicable?

Reviewer #1: Yes

Reviewer #2: Yes

4. Have the authors described where all data underlying the findings will be made available when the study is complete?

Reviewer #1: Yes

Reviewer #2: Yes

5. Is the manuscript presented in an intelligible fashion and written in standard English?

Reviewer #1: Yes

Reviewer #2: Yes

6. Review Comments to the Author

You may also provide optional suggestions and comments to authors that they might find helpful in planning their study.

Reviewer #1: The authors have addressed all of my earlier concerns. I have no further comments to raise and am satisfied with the current draft.

Reviewer #2: The study “Nutritional approach based on self-compassion versus energy-restricted diet approach in body dissatisfaction and disordered eating in adult women: a protocol for a randomized clinical trial” aimed to analyze the effects of a nutritional approach involving techniques that promote self-compassion compared to a traditional approach in women experiencing body image dissatisfaction, dietary restriction, and dysfunctional eating habits. I believe the article addresses an important and well-structured topic

7. PLOS authors have the option to publish the peer review history of their article (what does this mean? ). If published, this will include your full peer review and any attached files.

**Do you want your identity to be public for this peer review?** For information about this choice, including consent withdrawal, please see our Privacy Policy .

Reviewer #1: No

Reviewer #2: **Yes: ** Gabriel Fassina Ladeia

---

## [Editor Report · Acceptance letter]

PONE-D-24-40704R1

PLOS ONE

Dear Dr. de Souza,

I'm pleased to inform you that your manuscript has been deemed suitable for publication in PLOS ONE. Congratulations! Your manuscript is now being handed over to our production team.

Kind regards,

on behalf of

Dr. Leonardo Vidal Andreato

Academic Editor

PLOS ONE